# The B22 Dilemma: Structural Basis for Conformational Differences in Proinsulin B-Chain Arg22 Mutants

**DOI:** 10.3390/biom15040577

**Published:** 2025-04-12

**Authors:** Srivastav Ranganathan, Anoop Arunagiri

**Affiliations:** 1Max Planck Institute for Physics of the Complex Systems, 01187 Dresden, Germany; 2Department of Biological Sciences, East Tennessee State University, Johnson City, TN 37604, USA

**Keywords:** proinsulin simulation, diabetes, molecular dynamics, metadynamics

## Abstract

Proinsulin has three distinct regions: the well-folded A- and B-chains and the dynamic disordered C-peptide. The highly conserved B-chain is a hotspot for diabetes-associated mutations, including the severe loss-of-function R(B22)Q mutation linked to childhood-onset diabetes. Here, we explore R(B22)’s role in proinsulin stability using AlphaFold-predicted structures and metadynamics simulations to achieve enhanced sampling of the free energy landscape. Our results show that R(B22) stabilizes proinsulin by interacting with N86. Substituting R(B22) with E or Q disrupts this interaction, increasing conformational flexibility. The R(B22)Q variant exhibits a flattened free energy landscape, favoring unfolded states. Additional substitutions, including Gly, Ala, Lys, Tyr, Asp, and Phe, destabilize proinsulin to varying extents by weakening hydrogen bonding. Disrupting the R(B22)–N86 interaction broadly reduces inter-chain contacts, raising the risk of aggregation-prone states. Given the link between R(B22) mutations and diabetes, our study provides crucial molecular insights into proinsulin instability. These findings highlight the role of key inter-domain (A-Chain–B-chain, B-Chain–C-peptide, and A-Chain–C-peptide) interactions in maintaining protein structures and the implications this has for understanding disease-associated proinsulin variants.

## 1. Introduction

The intricate regulation of glucose homeostasis depends on the precise orchestration of insulin production and secretion by pancreatic beta cells [1]. Disruptions in this tightly controlled process can lead to diabetes mellitus, a complex metabolic disorder affecting millions worldwide. While diabetes can arise from a multitude of factors, including autoimmune destruction of beta cells (Type 1) [2] and peripheral insulin resistance (Type 2) [3], monogenic forms, resulting from single-gene mutations [4], offer invaluable insights into the fundamental mechanisms governing beta-cell function and insulin processing.

Mutations within the insulin gene (INS) have emerged as critical determinants of proper insulin function and glucose homeostasis. This work focuses on position B22 (corresponding to residue 46 in proinsulin) located within the insulin B-chain that plays a pivotal role in insulin’s interaction with its receptor and the intricate process of proinsulin folding and maturation. The significance of this residue is underscored by its high degree of evolutionary conservation and its involvement in stabilizing the insulin molecule through critical hydrogen bonds and electrostatic interactions [5]. Prior studies have demonstrated that amino acid substitutions at the B22 position can lead to a spectrum of diabetes phenotypes, ranging from mild forms of Maturity-Onset Diabetes of the Young (MODY) to severe insulin-deficient neonatal diabetes [5,6,7].

The R(B22)Q mutation, for example, has been identified in patients with MODY and even misdiagnosed as autoantibody-negative type 1 diabetes, highlighting the diagnostic challenges and phenotypic variability associated with INS mutations [5,6]. This mutation exhibits a dominant-negative effect, leading to progressive diabetes often diagnosed in adolescence and requiring insulin therapy [6]. At the cellular level, R(B22)Q results in partial proinsulin misfolding, with a significant portion retained within the endoplasmic reticulum (ER) and interfering with the processing of wild-type proinsulin [6,7]. While reducing the binding affinity of insulin to its receptor, this mutation does not induce significant ER stress, suggesting a distinct mechanism of beta-cell dysfunction [6]. Essentially, the R(B22)Q mutation produces a partially functional insulin molecule that, despite the presence of normal insulin from the other allele, is sufficient to disrupt glucose homeostasis and cause clinically significant diabetes [6].

In contrast, the R(B22)E mutation, investigated in a mouse model, reveals a strong gene–environment interaction [8]. Heterozygous mice with the R(B22)E mutation do not develop overt diabetes under normal conditions but readily develop the disease when challenged with a high-fat diet [8]. This suggests that R(B22)E creates a subthreshold predisposition to proinsulin misfolding, requiring an environmental ‘second hit’ to trigger clinical diabetes [8]. The severity of the R(B22)E phenotype also increases with gene dosage, with homozygous mice developing diabetes even on a normal diet, indicating that increasing levels of the mutant protein overwhelm the ER’s capacity to correctly process proinsulin [8].

Studies showing that even subtle changes to the ER’s redox environment can drastically change the folding outcomes of mutants at the B22 residue further emphasize its importance [9]. This highlights that the proper folding of proinsulin at this site is highly sensitive to the cellular environment and that even small mispairings or misfolding can have large consequences.

These striking phenotypic differences between R(B22)Q and R(B22)E, despite affecting the same position within the proinsulin molecule, raise a fundamental question: What is the structural basis for these distinct effects? How can single amino acid substitutions at the same location, differing only in their side chains, lead to such different outcomes in terms of proinsulin folding, processing, insulin secretion, and ultimately, glucose homeostasis?

To address this critical question, we combined in silico structure prediction with enhanced sampling simulations to study the conformational landscape of wild-type, R(B22)E, and R(B22)Q proinsulin. Using AlphaFold [10], we generated initial three-dimensional structures for each proinsulin variant. We then employed metadynamics to investigate the dynamics and free energies associated with different conformations. Our analysis focused specifically on the impact of the B22(Arg) to Gln or Glu substitutions on the proinsulin B-chain, examining how these changes alter the protein’s conformational ensemble and potentially influence intramolecular interactions. Our analyses reveal that Arg22 (R(B22)) stabilizes proinsulin by mediating a conserved inter-chain interaction with Asn86 (N(A21)), disruption of which (by amino acid substitutions) leads to increased conformational entropy and a flattened free energy landscape. R(B22) substitutions variably destabilize the native state and reduce inter-domain contacts, likely predisposing the molecule to misfold. Our study is an attempt to identify structural determinants underlying the distinct functional consequences of these two seemingly similar mutations, providing crucial insights into the molecular mechanisms driving beta-cell dysfunction in these forms of monogenic diabetes.

## 2. Materials and Methods

We employed AlphaFold 3.0 multimer [10] to predict the folded conformations of WT-proinsulin and the 8 R(B22)-substituted variants—R(B22)E, R(B22)Q, R(B22)A, R(B22)G, R(B22)F, R(B22)K, R(B22)Y, and R(B22)D. The following sequences were used as input for the AlphaFold structure prediction:

**WT:** FVNQHLCGSHLVEALYLVCGERGFFYTPKTRREAEDLQVGQVELGGGPGAGSLQPLALEGSLQKRGIVEQCCTSICSLYQLENYCN

**R(B22)E:** FVNQHLCGSHLVEALYLVCGEEGFFYTPKTRREAEDLQVGQVELGGGPGAGSLQPLALEGSLQKRGIVEQCCTSICSLYQLENYCN

**R(B22)Q:** FVNQHLCGSHLVEALYLVCGEQGFFYTPKTRREAEDLQVGQVELGGGPGAGSLQPLALEGSLQKRGIVEQCCTSICSLYQLENYCN

**R(B22)A:** FVNQHLCGSHLVEALYLVCGEAGFFYTPKTRREAEDLQVGQVELGGGPGAGSLQPLALEGSLQKRGIVEQCCTSICSLYQLENYCN

**R(B22)G:** FVNQHLCGSHLVEALYLVCGEGGFFYTPKTRREAEDLQVGQVELGGGPGAGSLQPLALEGSLQKRGIVEQCCTSICSLYQLENYCN

**R(B22)F:** FVNQHLCGSHLVEALYLVCGEFGFFYTPKTRREAEDLQVGQVELGGGPGAGSLQPLALEGSLQKRGIVEQCCTSICSLYQLENYCN

**R(B22)K:** FVNQHLCGSHLVEALYLVCGEKGFFYTPKTRREAEDLQVGQVELGGGPGAGSLQPLALEGSLQKRGIVEQCCTSICSLYQLENYCN

**R(B22)Y:** FVNQHLCGSHLVEALYLVCGEYGFFYTPKTRREAEDLQVGQVELGGGPGAGSLQPLALEGSLQKRGIVEQCCTSICSLYQLENYCN

**R(B22)D:** FVNQHLCGSHLVEALYLVCGEDGFFYTPKTRREAEDLQVGQVELGGGPGAGSLQPLALEGSLQKRGIVEQCCTSICSLYQLENYCN

The primary sequence of wild-type proinsulin provided in Figure 1A indicates the position of R(B22).

The structural models from AlphaFold 3.0 can be downloaded from the GitHub repository—https://github.com/sranga88/Proinsulin-R-B22--AF3-Models.git (accessed on 27 March 2025). The C-peptide region of proinsulin is highly dynamic, however, and is predicted with low confidence by AlphaFold. The dynamics of the protein therefore becomes crucial for its function.

To establish that the AlphaFold-predicted structures indeed correspond to a stable, minimum energy conformation and to identify other conformations accessed by the protein, we performed metadynamics simulations. Metadynamics is an enhanced sampling approach that is employed to construct a free energy landscape of proteins along a chosen collective variable [11,12]. Here, we employ a history-dependent biasing potential to ensure an efficient sampling of all metastates (folded and partially folded conformations). To explore the conformational landscape associated with chain compaction, well-tempered metadynamics simulations were performed using the Colvars module implemented in NAMD3 [13]. The radius of gyration (Rg) of the protein backbone (residues 1–86) was chosen as the collective variable (colvar) to quantify the global compactness of the polypeptide. Specifically, the Rg was computed using the Cα atoms (atom name “CA”) of residues 1 through 86. The collective variable was bounded by harmonic wall potentials to confine sampling within a physically meaningful range of values. Soft harmonic walls were applied between 5 Å and 20 Å, with lower and upper wall constants set to 2.0 kcal/mol/Å^2^ and 3.0 kcal/mol/Å^2^, respectively. This prevented unphysical expansion or over-compaction during the enhanced sampling. This setup enabled efficient exploration of conformational states across a broad range of Rg values allowing us to sample a range of compact (lower Rg values ->12 Angstroms) and open conformations (Rg > 14 Å) of proinsulin.

Metadynamics was conducted in the well-tempered formulation, which tempers the height of successive hills to promote smooth convergence of the free energy surface. Hills were deposited every 500 steps, with a hill width of 1 Å, a hill height (initial bias) of 0.3 kcal/mol, and a bias temperature of 1590 K, corresponding to moderate biasing strength. The colvars and bias potentials were output every 500 steps, and the accumulated bias was written in the form of a free energy file and hills trajectory for post-processing and visualization. A 200 ns metadynamics trajectory was used to compute the potential mean force (PMFs) along the reaction coordinate. The system sampled all states along the collective variable within the simulation timescale and the free energy profiles showed convergence. Appendix A shows the convergence of metadynamics hill heights. The energy deposited (in negative) corresponding to each state is the free energy of being in that state.

All simulations were carried out using NAMD2.3 [13] with the CHARMM36m forcefield [14], and associated parameter sets. The simulated system consisted of the protein solvated in an explicit water box with periodic boundary conditions applied in all directions. The simulation temperature was set to 310 K throughout, reflecting physiological conditions. Non-bonded interactions were treated using a 12 Å cutoff with a switching function starting at 10 Å, and a pairlist distance of 13.5 Å. Long-range electrostatics were computed using the Particle Mesh Ewald (PME) method with a grid spacing of 1.0 Å. All hydrogen-containing bonds were constrained using the rigidBonds setting in NAMD, enabling a time step of 2 fs. Non-bonded interactions were evaluated every step, while full electrostatic calculations were performed every 2 steps. Temperature control was maintained using Langevin dynamics with a damping coefficient of 3 ps^−1^, applied to non-hydrogen atoms only. Pressure control was achieved using the Langevin piston method, targeting 1 atm with a piston period of 100 fs and decay time of 50 fs, matching the simulation temperature of 310 K. A constant volume (NPT) ensemble was maintained without using a flexible cell, appropriate for isotropic water boxes. The simulation was run for a total of 200,000,000 steps, corresponding to 200 ns of simulation time. Metadynamics simulations were run till the PMFs exhibited convergence. To compute hydrogen bonds and the h-bond maps, we use structures that correspond to the free energy minima in Figure 1. The structures were visualized using VMD [15]. Our choice of simulation parameters is similar to previous molecular dynamics studies involving proteins [16,17,18,19,20].

## 3. Results

The full-length proinsulin is 86 residues long wherein residue numbers 1 to 29 correspond to the B-chain, 30–65 correspond to the C-peptide and 66–86 correspond to the A-chain (see Figure 1A). The nomenclature style R(B22) or Arg22 refers to the 22nd residue in the B chain, which corresponds to an Arg in the 46th position in the full-length protein. Hereon, unless otherwise stated, we refer to the name of the variants such as R(B22)Q, R(B22)E, etc., following this nomenclature. We performed atomistic resolution simulations of WT proinsulin and eight R(B22) substituted variants of the protein to study how a disruption to the R(B22)-mediated interaction network could influence the conformational landscape and the stability of proinsulin.

### 3.1. WT Proinsulin Shows a Stable Minimum Around Compact Conformations

To enable efficient sampling of the conformational landscape, we performed metadynamics simulations (see Section 2) of the full-length WT and mutant proinsulin structures generated by AlphaFold3. In Figure 1B–D, we show the ensemble-averaged 3D structures of the proinsulin WT and R(B22) variants obtained from metadynamics simulations. As evident from the free energy profiles, the 3D structure of R(B22)Q is less compact and has fewer stable hydrogen bonds, as seen from the fewer green-colored spheres in Figure 1D as compared to Figure 1B,C. In Figure 2, we show the free energy profiles for WT proinsulin (Figure 2A) computed from the metadynamics simulation runs and compare it to the corresponding profiles for R(B22)-substituted proinsulins with known phenotypes, R(B22)E (Figure 2B) and R(B22)Q (Figure 2C). As expected, the WT-proinsulin shows a deep, sharp global minimum at an Rg value of ~12 Å, with no other subpopulations or local minima in the free energy landscape (Figure 2A) suggesting a stable native conformation. As we introduce an “E” substitution at the site of R(B22), we see a relatively lower stability of the native conformations (~12 Å) and additional “open” conformations corresponding to a local minimum of around 16–18 Å. The protein still predominantly samples a native-like conformation in the case of the R(B22)E variant (Figure 2B). The R(B22)Q variant, on the other hand, shows dramatically different behavior, with the native conformation showing complete destabilization. As opposed to Figure 2A,B, the free energy landscape for R(B22)Q is flatter, with no deep minima, suggesting that the unfolded/partially folded states of the protein are accessed with a high likelihood compared to the WT and R(B22)E mutant. We then analyze the source of the structural deviation by superimposing the ensemble-averaged structures corresponding to the free-energy minima in Figure 2. As evident from the superimposed structures, the WT and R(B22)E variant show significant overlap in the structured regions (A- and B-chains) of the protein, while the C-peptide shows the strongest deviation (Figure 3A,B) and signatures of structure in the variants (Figure 3A–C) compared to the mostly unstructured C-peptide in the WT (Figure 3A,B). Overall, the results suggest that a modification to the R(B22) residue results in structural deviations mediated by the C-peptide of proinsulin.

### 3.2. Disruption of Key A-Chain–B-Chain Interaction Leads to a Disruption of C-Peptide-Mediated Interactions

To further understand the molecular basis for the relatively lower stability of the native state in the variants compared to the WT, we first computed the number of stable inter-chain hydrogen bonds in the WT with that of R(B22)E and R(B22)Q. The WT shows the highest degree of A-Chain–B-chain, and A-Chain–C-peptide interactions, which result in a more compact structure compared to the R(B22)E and R(B22)Q (Figure 4A,B). On the other hand, while the R(B22)Q shows fewer inter-chain interactions across all three categories in Figure 4, R(B22)E compensates for the fewer A–B and A–C interactions with an increase in B-Chain–C-peptide interactions possibly stabilizing the more compact conformations that correspond to WT-like dynamics. To further understand how substitutions at the site of R(B22) reshape the conformational landscape of the protein, we plot a map of stable hydrogen-bond interactions in proinsulin and the R(B22)Q/E variants. In Figure 5, large bright-colored circles correspond to bonds that are found in a large % of structures that are part of the ensemble. Further, contacts near the diagonal indicate local interactions that are nearby in the primary sequence while the off-diagonal elements indicate more long-range interactions across the A-Chain, B-Chain, or C-peptide. Crucially, R(B22) is involved in a key stable interaction with N86, thereby pinning the A-chain and B-chains together. The A-Chain–B-chain interaction is further stabilized (Figure 5A, blue box) in the WT by other interactions involving residues in the proximity of R(B22) and N86 (See Figure 1A for primary sequence location). These include T30-N83, F25-Y84, and L6-C71. The net effect of these interactions (contacts within the blue box in Figure 5A) is a stable compaction that is an outcome of the A-chain and B-chain being close together in the three-dimensional space. Similarly, C-peptide and A-chain interact via Y84-R65 and S77-L58 as the primary contacts along with a set of dynamic interactions that form and break constantly (smaller, darker circles in Figure 5A, red box). Overall, the prevalence of several inter-chain interactions between A-chain–B-chain and A-chain–C-peptide helps stabilize the WT proinsulin in the more compact conformations (Figure 1B–D as well as Figure 2).

As we substitute the R(B22) to E, we observe a direct impact on the stable interactions between the A-Chain–B-chain. The R(B22)E substitution not only disrupts the key R(B22)–N86 interaction, which characterizes the WT-proinsulin (Figure 5A), but also results in a complete abrogation of the adjoining T30–N83 interaction. It is, however, compensated for by a new stable interaction between K29–E69, while the F24–Y84 and L6–C71 interactions from the WT are still retained, thereby ensuring that the A-Chain–B-Chain interactions continue to remain stable. The A-Chain–C-peptide interactions, on the other hand, instead of being characterized by a few strong interactions in the WT, are stabilized by a cluster of several weaker interactions in R(B22)E. The R(B22)E protein is also stabilized by K29–E33 interactions between the edge of the B-chain and the C-peptides, a feature not observed in the WT. E33 is also involved in an interaction with T73, a C-peptide–B-chain interaction, thereby making this residue a key contributor to the compactness of the protein. The net effect is the ability of the R(B22)E protein to access more open conformations (Figure 1C and Figure 2B) while the more compact, native-like conformations continue to remain stable.

Unlike the R(B22)E variant, an R(B22)Q substitution results in a dramatic abrogation of not only the A-chain–B-chain contacts (blue box in Figure 5C) but also a complete destabilization of A-chain–C-peptide interactions (red box in Figure 5C). This results in a complete unraveling of the overall fold of the protein due to a breakdown of these long-range, inter-domain interactions giving us a flatter landscape and open conformations (Figure 1D and Figure 2C). The local secondary structural features, however, continue to persist in R(B22)Q as seen from the near-diagonal contacts in Figure 5C. Metadynamics simulations of the WT-proinsulin and the R(B22)E and R(B22)Q substituted mutants reveal a key feature of the WT structure. The R(B22)–N86 interaction has been found in several different proinsulin structures corresponding to different species [21,22], and is a vital part of establishing other stable interactions in the vicinity of R(B22)–N86 as well as interactions that stabilize A-chain–C-peptide interactions in the protein (Figure 5).

### 3.3. Non Q/E, R(B22)-Variants Display Increased Conformational Flexibility

Substitutions to the R(B22) therefore either result in a partial (Figure 1C and Figure 2B) or complete loss of stability (Figure 1D and Figure 2C) of the compact, native conformations of proinsulin. The conformational landscape of the protein is, therefore, highly sensitive to substitutions at this location, as evident from the loss of function in R(B22)Q mutants [6,9]. Consistent with these results, a comprehensive sequence alignment study by Landreh, et al. [23] also reveals that R(B22) is a site that shows >70% residue identity, in a highly conserved B-chain of proinsulin. Therefore, in addition to the two variants with known phenotypes, R(B22)E and R(B22)Q, we further introduce other substitutions to R(B22) to understand how they could influence the conformational landscape of the proinsulin. Interestingly (Figure 6A–F), in all variants under study, we observe a destabilization of the compact conformation compared to the WT (Figure 1B and Figure 6) and an increased likelihood of the protein accessing more open conformations (Figure 1B and Figure 6). The extent of destabilization of the WT, however, varies depending on the identity of the substituted residue. While alanine substitution results in an increased likelihood of sampling open conformations (Figure 6 and Figure 7), the WT is still the most predominant state of the protein. On the other hand, G, K, F, and Y variants show a higher likelihood of sampling the open conformations. Interestingly, R(B22)D shows a WT-like behavior with a strong preference for compact native-like conformations. Interestingly, D-substitution is among the few substitutions evolutionarily observed at this location [23]. In Figure 7, we also show a more detailed look at open, and closed conformations of the different variants under study showing how the conformational variations are largely mediated by the C-peptide of the protein, with a reduction in A–B and A–C interactions resulting in more open conformations (Figure 8) Interestingly, all the R(B22)-substituted variants in Figure 6 show a significant reduction in stable A-chain–B-chain and A-chain–C-peptide interactions compared to the WT proinsulin (Figure 8A,B).

### 3.4. Non-Q/E, R(B22)-Variants Display Consistent Disruption in A-Chain–B-Chain Interactions Compared to WT

On the other hand, the WT shows the weakest B-chain–C-peptide interactions. The variants, except R(B22)Q, partially compensate for the loss of A-chain–C-peptide and A-chain–B-chain interactions via increased B-chain–C-peptide interactions preventing a complete unraveling of the protein (Figure 8C). This is further evident from the sparseness of strong interaction signals in the blue- and red-boxed regions in Figure 9, compared to the same for WT-proinsulin (Figure 5A). Overall, our results suggest that a disruption in the R(B22)–N86 interaction results in destabilization of the A-Chain–B-chain interaction network, which subsequently also results in more flexible A-chain–C-peptide interactions, and therefore a more conformationally flexible protein that can access more open conformations more readily. Even a partial destabilization or access to the open states could result in dramatic ramifications for protein processing, storage, and downstream function. For instance, the loss of native long-range interactions could result in an increased likelihood of inter-protein interactions making the substituents more aggregation-prone.

## 4. Discussion

Mutations within the INS gene, the root cause of MIDY, initiate a cascade of detrimental cellular events. These mutations directly interfere with proinsulin folding (within the ER), and thereby its ER-to-Golgi trafficking [24,25,26]. This disruption leads to a progressive decline in insulin production and, thus beta-cell function, ultimately culminating in insulin deficiency and the onset of hyperglycemia. While both A-chain and B-chain mutations are implicated in MIDY, they often manifest with distinct clinical profiles.

Mutations in the insulin A-chain often result in severe phenotypes, notably neonatal diabetes. For instance, the C(A7)Y (Akita) mutation, which substitutes cysteine with tyrosine, disrupts a critical disulfide bond. This disruption leads to significant proinsulin misfolding, increased ER stress, and a strong dominant-negative effect on wild-type proinsulin [25,27]. Another A-chain mutation, E(A4)K, also impacts disulfide bond formation. However, biochemical studies reveal that removing the A6–A11 disulfide bond can rescue the mutant’s trafficking and folding, highlighting the specific roles of individual disulfide bonds in proinsulin [24]. In contrast, the L(A16)P mutation, induces severe proinsulin misfolding and ER retention [24]. This misfolding, however, cannot be rescued by eliminating the A6–A11 disulfide bond, suggesting a distinct mechanism of ER retention compared to E(A4)K.

Mutations within the insulin B-chain, such as G(B20)R and P(B28)L, generally lead to milder forms of diabetes [28]. These mutations cause partial proinsulin folding defects and less severe ER export defects. Notably, clinical presentations and disease severity vary among patients with these B-chain mutations. For instance, a patient with the G(B20)R mutation exhibited mild, diet-controlled diabetes, whereas a patient with the P(B28)L mutation presented with classic diabetes symptoms and a strong family history [28]. Other B-chain mutations, including H(B5)D, V(B18)A, G(B8)V, and R(B22)Q, also contribute to MIDY, but with differing degrees of severity. The V(B18)A mutation appears to be better tolerated, while the R(B22)Q mutation typically results in diabetes onset during adolescence [6,24]. Interestingly, the V(B18)A mutation allows for subtle anterograde proinsulin trafficking and even some insulin secretion [24]. Similar to the E(A4)K A-chain mutation, the H(B5)D B-chain mutation can be rescued by removing the A6–A11 disulfide bond. This observation underscores the complex interplay of disulfide bonds in regulating proinsulin folding and trafficking and the resulting variability in clinical outcomes.

The fundamental pathophysiology of MIDY revolves around ER stress, the dominant-negative interference with wild-type proinsulin, the disruption of critical disulfide bonds vital for protein stability, and the formation of aberrant proinsulin complexes that further impede cellular function [24,29]. As mentioned earlier, MIDY mutations target proinsulin folding and intracellular trafficking, directly compromising insulin production. A thorough understanding of these genotype–phenotype correlations is paramount for accurate clinical management, informed genetic counseling, and the development of targeted therapeutic strategies aimed at mitigating the devastating effects of MIDY. Given the profound impact of these mutations on proinsulin folding and the resulting clinical heterogeneity, a molecular-level understanding of how mutations impact the stability and conformational behavior of proinsulin is important to understanding the molecular basis of mutation-induced diabetes. The folding of proinsulin is contingent on three disulfide bonds (B19–A20, B7–A7, and A6–A11 (see Figure 1A)), and perturbations to the folding landscape could result in kinetic traps and misfolding/aggregation events [30]. In this context, the highly invariant nature of the proinsulin A- and B-chains across billions of years of evolution [23] suggests that these two domains have evolved under a strict evolutionary constraint [30]. The extreme folding stress that the proinsulin sequence is under has been previously established using an F(B24) mutation, which shows impaired secretion upon an F->Y mutation, an otherwise frequent amino acid swap in nature [31]. This suggests that each residue in the proinsulin B-chain has evolved to the “edge of foldability” [30]. In other words, the B-chain sequence is under such tight evolutionary control that even slight alterations could result in dramatically different free energy landscapes, therefore resulting in misfolding and/or aggregation [30].

Phenotypically, mutations at diverse locations on the proinsulin B-chain have resulted in aberrant localization [6,32] leading to loss of function. Among the several B-chain mutations associated with diabetes is the R(B22)Q mutation, an early childhood mutation. R(B22), located in the B-chain of proinsulin, also shows a very low sequence variability across species, with the site itself showing strong conservation signals across proinsulin sequences from mammals, birds, amphibians, and fish [23]. Charged residues, owing to their ability to participate in key electrostatic interactions such as salt bridges [33,34], often contribute significantly to protein stability [35]. Substitutions to highly conserved, charged amino acid residues could help us unravel the fundamental molecular forces behind protein stability. Similarly, the arginine at site 22 in the proinsulin primary structure (B22) shows a general interaction trend with the carboxylate of Asn at location 86 (A21) in several structures. However, a bottom-up understanding of how this R(B22) contributes to the overall stability of proinsulin is not known.

Using AlphaFold-predicted structures and metadynamics simulations, we demonstrate that R(B22) plays a crucial role in maintaining the compact native conformation of proinsulin through its interaction with N86. The free energy landscapes and hydrogen-bond analyses highlight the extent to which substitutions disrupt this interaction network and alter the folding stability of proinsulin. The wild-type (WT) proinsulin exhibits a well-defined global minimum at a radius of gyration (Rg) of ~12 Å, indicative of a stable, native conformation. In contrast, substitutions at R(B22) result in varying degrees of destabilization. Specifically, the R(B22)E mutant exhibits an additional local minimum around Rg ~17 Å, indicating the presence of more open, less-folded states, but it still predominantly retains a WT-like conformation. On the other hand, the R(B22)Q substitution leads to a significant flattening of the free energy landscape, suggesting a loss of conformational stability and an increased probability of sampling unfolded or partially folded states. Hydrogen-bond network analysis reveals that while the WT maintains a dense interaction pattern between the A-chain, B-chain, and C-peptide, the R(B22)Q mutation disrupts these inter-chain interactions entirely, leading to a highly flexible and destabilized structure. Interestingly, the R(B22)E variant, while also destabilizing the A-chain–B-chain interaction, partially compensates by enhancing B-chain–C-peptide interactions. This compensation may explain why R(B22)E mutants retain some degree of structural stability compared to R(B22)Q. However, both variants show a clear reduction in stable inter-chain contacts, which could have significant implications for proinsulin folding and aggregation. These results are consistent with experimental evidence [6], which reveals that the R(B22)Q results in a severe loss of function associated with childhood diabetes [5,6] while the R(B22)E is a milder mutation in comparison [8].

To generalize these findings, we extended our study to additional R(B22) substitutions, including G, A, K, Y, F, and D. All substitutions led to a shift in the free energy landscape toward more open conformations, with varying degrees of destabilization. While A and D substitutions still allowed some WT-like conformations to persist, variants such as G, K, F, and Y exhibited a higher tendency to adopt expanded states. Our results further reiterate that the proinsulin conformational landscape is highly susceptible to substitutions in B22 as it is to changes in B24 [31], and the resultant access to more open conformation could influence the oxidative refolding of proinsulin and/or the correct disulfide pairing.

A detailed hydrogen-bond map comparison for R(B22) variants confirms that the disruption of the R(B22)–N86 interaction leads to a broader destabilization of inter-chain contacts. While most variants compensate for this loss by increasing B-chain–C-peptide interactions, the overall effect is a loss of long-range inter-domain stability, leading to a more conformationally flexible protein. Given the known association of R(B22) mutations with diabetes phenotypes, these findings provide crucial insights into the molecular mechanisms by which single amino acid changes at this site can significantly impact proinsulin folding and function. In conclusion, our study establishes R(B22) as a key stabilizing residue in proinsulin, with its substitution leading to varying degrees of destabilization. The loss of stable inter-chain interactions and the increased sampling of open states could have downstream consequences, including impaired proinsulin processing, increased aggregation propensity, and potential functional deficiencies. These findings offer a mechanistic framework for understanding the molecular basis of disease-associated R(B22) mutations and highlight the importance of electrostatic interactions in maintaining protein stability.

While the current study employs extensive single-molecule simulations that suggest significant conformational differences, the molecular implications of these structural deviations in downstream aggregation are not fully addressed. To further understand how the huge perturbation in the conformational landscape can influence proinsulin aggregation and/or misfolding, self-association simulations with the protein need to be performed. While the current study shows significant alterations to the hydrogen-bond map in R(B22) variants, a more detailed study with additional modifications is required to understand how the perturbations manifest themselves in the altered free energy profiles. Despite existing cell and animal model studies on B(22)E and B(22)Q [6,8], our work offers new structural insights into proinsulin folding and identifies key amino acids influencing this process, necessitating further investigation. We recognize the absence of in vitro studies as a key limitation and will prioritize these in subsequent research.

## 5. Conclusions and Future Directions

Understanding proinsulin misfolding due to INS gene mutations is crucial for developing therapies. Potential strategies include CRISPR/Cas9 gene editing and targeting free thiols to prevent aberrant disulfide bond formation [24,36]. The high conservation and sensitivity of the R22 (B22) residue in proinsulin, particularly its interaction with N86 (A21), presents a potential therapeutic target. Our simulations demonstrate that disrupting this R22–N86 interaction destabilizes critical proinsulin interactions, leading to more open, “aggregation-prone” conformations. Given that R22 is highly conserved and mutations at this site are linked to diabetes, therapeutic strategies aimed at stabilizing or restoring the R22–N86 interaction could mitigate proinsulin misfolding and aggregation. This could involve small molecules designed to reinforce this interaction, or gene therapies targeting the restoration of proper R22 function. Furthermore, the destabilization of A-chain–C-peptide and B-chain–C-peptide interactions provides additional targets for therapeutic interventions, as stabilizing these interactions could also prevent proinsulin misfolding, ultimately preserving insulin production and secretion.

## Figures and Tables

**Figure 1 biomolecules-15-00577-f001:**
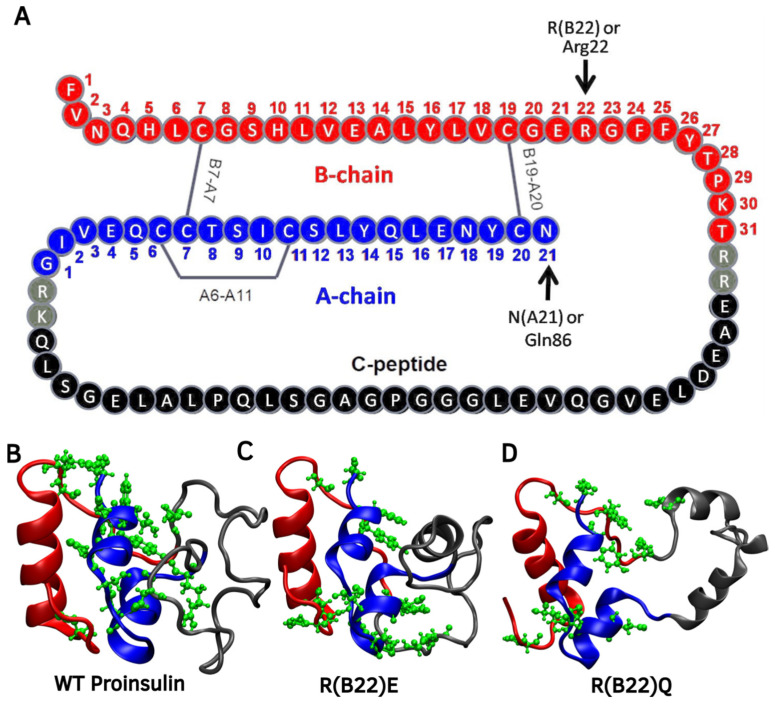
Proinsulin Domain Organization: (**A**) The primary structure of proinsulin shows individual chains in different colors and the position of R22 and N86 (discussed later). The ensemble-averaged structures corresponding to the free-energy minima from metadynamics simulations, for (**B**) WT, (**C**) R(B22)E, and (**D**) R(B22)Q mutants. The residues that show the highest involvement in hydrogen bonds are marked in green CPK ball-stick representation. The B-chain is shaded in red, the A-chain in blue, and the C-peptide in gray.

**Figure 2 biomolecules-15-00577-f002:**
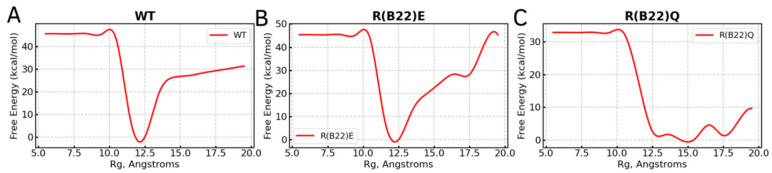
Free Energy profiles from metadynamics simulations for WT proinsulin (panel (**A**)) compared to that of (**B**) R(B22)E and (**C**) R(B22)Q mutants. The reaction coordinate on the x-axis is the radius of gyration of the protein chain, with large values indicating a less compact, and potentially less-folded protein. The wild-type proinsulin shows a clearly defined global minimum at Rg~12 Å. R(B22)E substitution results in reduced stability of the WT-like native minimum with a second, less-folded state being sampled at Rg~17 Å. The R(B22)Q substitution, on the other hand, shows a clear destabilization of the native state with a flatter free energy profile with multiple states being sampled. A defined minimum with a deep well suggests a stable conformation while a flatter landscape, like in panel (**C**), suggests a conformationally flexible protein with no stable minimum.

**Figure 3 biomolecules-15-00577-f003:**
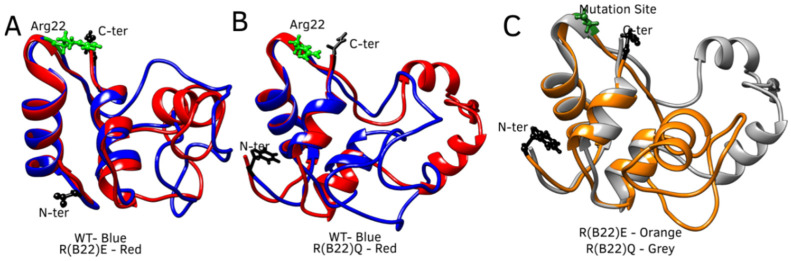
Conformational divergence of WT with respect to the R(B22) mutants. The ensemble-averaged free-energy minimum conformations of the WT (shown in blue ribbons) are superimposed onto the corresponding conformations of (**A**) R(B22)E (in red ribbons) and (**B**) R(B22)Q (in red ribbons). The C-peptide shows significant helical propensity in the mutants, compared to the WT, and shows a more open conformation in R(B22)Q, (**C**) structural comparison of R(B22)Q (in grey), and R(B22)E (in orange).

**Figure 4 biomolecules-15-00577-f004:**
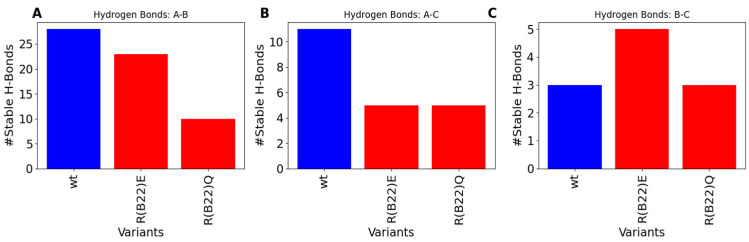
Comparison of stable inter-chain interactions across WT, R(B22)E and R(B22)Q: (**A**) Stable H-bonds across A- and B-chains, (**B**) between A-chain and C-peptide, and (**C**) between B-chain and C-peptide. The mutants show an overall reduction in the number of inter-chain interactions, with R(B22)Q showing the highest disruption in all 3 categories of interchain contacts.

**Figure 5 biomolecules-15-00577-f005:**
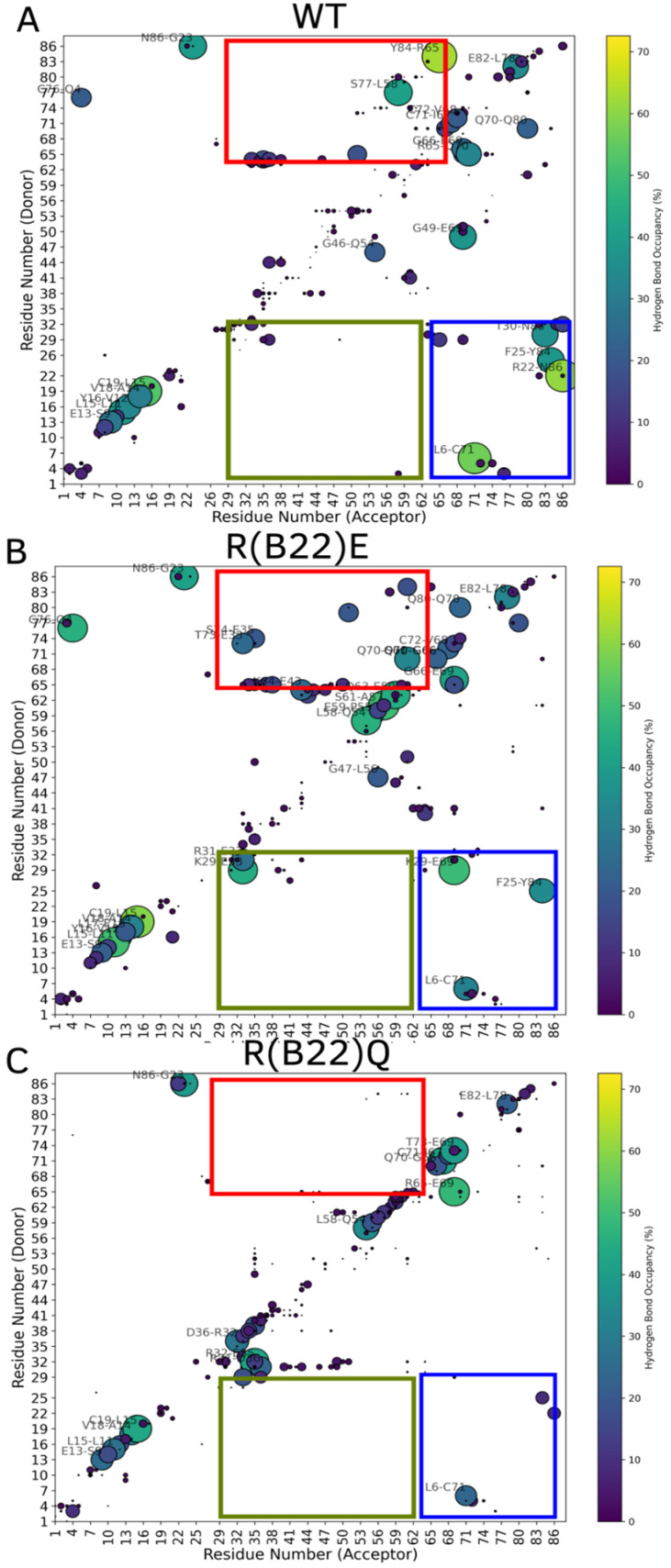
Hydrogen-bond maps for (**A**) WT, (**B**) R(B22)E, and (**C**) R(B22)Q showing the densest maps for the WT, with a reduction in interactions for the two R(B22)-substituted proteins in (**B**,**C**). The region bound by the blue box (A-chain–B-chain interactions) shows the highest degree of disruption due to the R(B22) substitution. R(B22)Q shows a complete abrogation of contacts while R(B22)E shows a less dense map in this region. A similar trend can also be seen for the region enclosed by the red box for visual aid (A-chain–C-peptide interactions). The interactions between B-chain and C-peptide residues are shown in the green box for visual aid indicating regions that show the highest difference in interactions across structures. The color and size of the circles are both indicative of the frequency of the interaction (occupancy) in a population of structures that correspond to the free energy minima in Figure 2. Interactions that are frequently observed are represented by brightly colored large circles while infrequent interactions are represented by small dark circles. We would like to note that while discussing hydrogen-bond maps, we refer to the absolute positions of the amino acids in the full-length proinsulin.

**Figure 6 biomolecules-15-00577-f006:**
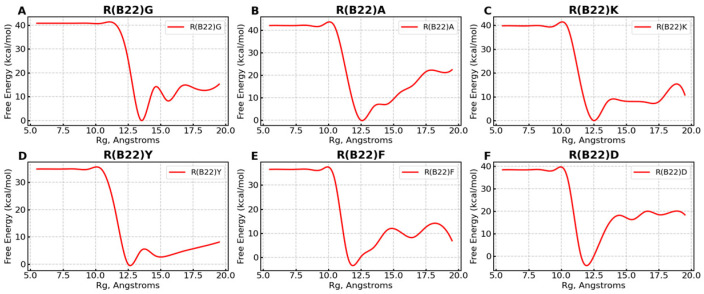
Free Energy profiles for non-E/Q substitutions at R(B22). The panels (**A**–**F**) correspond to G-, A-, K-, Y-, F-, and D-substituted proteins at residue 22. Note that all these substitutions result in a destabilization of the native-like state (Rg of ~12 Å) and a corresponding sampling of less-folded conformations with Rg > 14 Å.

**Figure 7 biomolecules-15-00577-f007:**
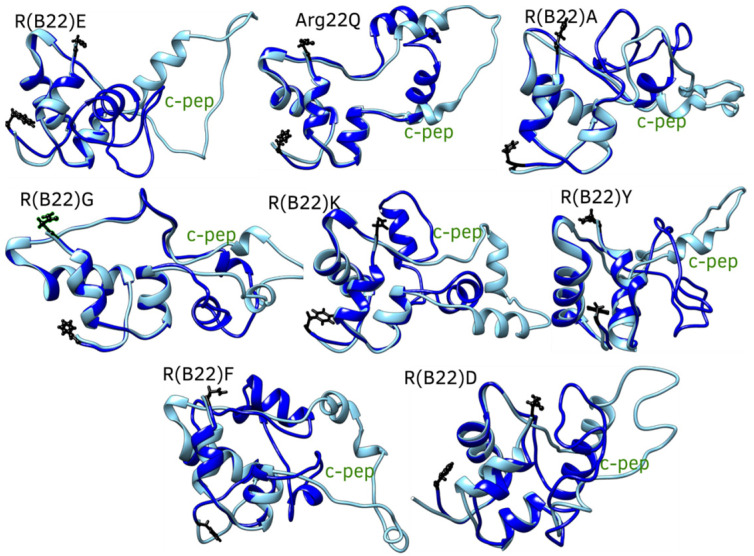
The comparison of the different R(B22) variants in their open (cyan) and closed (blue) conformations. Structures with Rg between 11 and 13 A in the free energy landscape were considered closed-state structures while those above 14 A were considered open-state conformations. The open-state conformations show a significant deviation in the c-peptide region of the protein.

**Figure 8 biomolecules-15-00577-f008:**
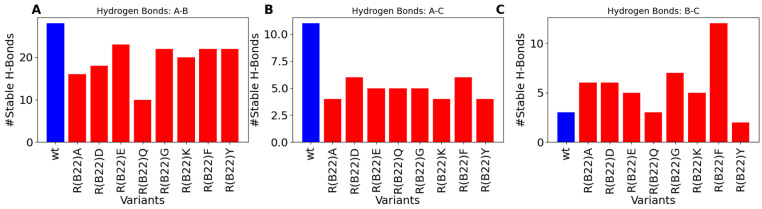
Comparison of stable inter-chain interactions across non-E/Q substituted proteins at the site of R(B22). Overall, we see a decrease in (**A**) A-chain–B-Chain and (**B**) A-chain–C-peptide, for all the substitutions under study. (**C**) The B-chain–C-peptide interactions, on the other hand, are either conserved or show an increase in all the substituted peptides (except R(B22)Y).

**Figure 9 biomolecules-15-00577-f009:**
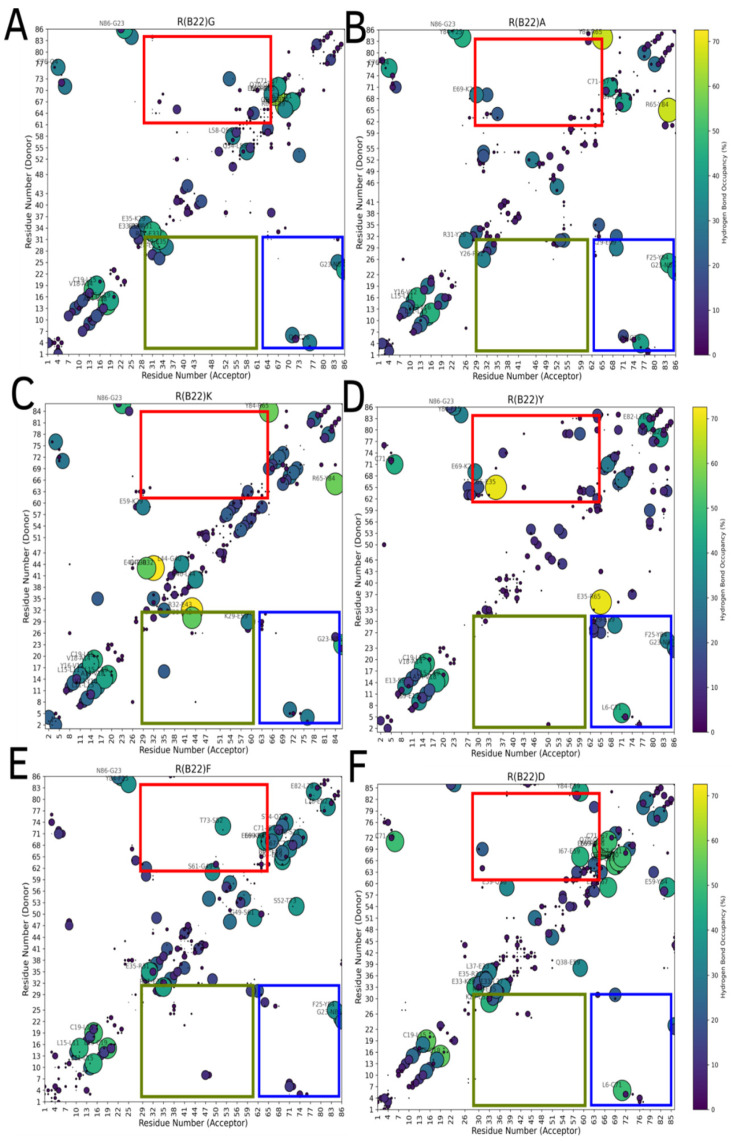
Hydrogen-bond maps for non-E/Q substituted variants at residue R(B22). (**A**), (**B**), (**C**), (**D**), (**E**), and (**F**) panels correspond to an ensemble-averaged H-bond map for R(B22)G, R(B22)A, R(B22)K, R(B22)Y, R(B22)F, and R(B22)D, respectively. Across all these variants, we see a reduction in A-Chain–B-chain contacts (region shaded blue) compared to the WT (refer to Figure 5A). The local hydrogen-bond network within these chains (close to the diagonal), however, is largely conserved across all substituents. The color and size of the circles are both indicative of the frequency of the interaction (occupancy) in a population of structures that correspond to the free energy minima in Figure 6. Interactions that are frequently observed are represented by brightly colored large circles while infrequent interactions are represented by small dark circles. We note that when discussing hydrogen-bond maps, we are referring to the absolute positions of the amino acids in the full-length proinsulin.

## Data Availability

All data are reported in the article. No unreported data have been discussed in the article.

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
