# Peer review of "The B22 Dilemma: Structural Basis for Conformational Differences in Proinsulin B-Chain Arg22 Mutants"

_biomolecules, 2025, doi:10.3390/biom15040577_

Round 1
Reviewer 1 Report
Comments and Suggestions for Authors
The authors investigated the role of Arg(B22) in proinsulin stability by AlphaFold predictions and metadynamic simulations. They showed that Arg(B22) maintains structure by interacting with Asn86, but replacement with Gln or Glu disrupts this interaction, increasing conformational flexibility. The R(B22)Q variant favors a more unfolded state, while other mutations destabilize proinsulin by weakening hydrogen bonds. This study explains at the molecular level how R(B22) mutations disrupt proinsulin structure and its association with diabetes. However, authors should make some corrections throughout the text.
- An appropriate form of citation style should be applied based on the biomolecules.
-
Please give a reference for lines 36 to 38.
-
“including Gly, Ala, Lys, Tyr and Phe” In this sentence, in line 16, the comma is in the wrong position. It should be before “and”.
- The sentence “The dynamics of the protein therefore become crucial to its function.” is on line 99 and 'therefore' should be separated by two commas; 'becomes' should be 'become' instead of 'becomes'.
- The sentence between lines 46 and 48 should be made more fluent.
- The sentence between lines 42 and 43 should be improved by adding “which was”.
- In line 79, Latin phrases such as "in silico" should be italicized.
- Some sentences contain extra spaces. Please revise them (e.g., line 81).
- In lines 95-97, writing the number as “eight” instead of “8” is more in line with academic writing rules, and it is more correct to use “:” instead of “ - ”.
- Lines 100-102 should be “AlphaFold-predicted” and “minimum-energy” should be combined.
- In lines 136-138, “observe” instead of “see” is more appropriate in academic writing.
- In lines 338-339, “a certain” instead of “some” sounds more academic.
- In line 344, the phrase "(CITE Some Experiment if available)" appears. Please remove it.
- In lines 356-358, “results in” is more precise than “leading to” and “flexible” is more common than “malleable”.
- For the abstract section, it would be helpful to make the methods more explicit. For example, why AlphaFold and metadynamics are preferred. Also, to emphasize the association with diabetes more strongly: “These findings provide important contributions to understanding the molecular mechanisms of diabetes-associated proinsulin mutations.”
- The introduction could better contextualize why R(B22) mutations are specifically important and previous studies on this topic.
- A comparison with other insulin mutations associated with diabetes could be included. More explanation on the pathophysiology of the mutations should be added.
- For the methodology part, it should be more clearly explained why the simulation parameters were chosen based on the literature.
- For the conclusion, more details on the statistical significance of the results should be provided. The question “How do the results differ from other previously identified insulin mutations?” should be answered.
- Figures (e.g., Fig. 1 and 2) need more explanatory discussion. The captions of the figures should be made more descriptive.
- For the discussion part, the implications of the findings for clinical applications should be discussed further. A topic such as “Do these mutations have the potential to be considered as therapeutic targets in the future?” could be addressed. Limitations of the study (e.g., lack of in vitro/in vivo validation) should be clearly stated.
- In the overall conclusion, the original contribution of the study should be clarified. A statement such as “This study sheds light on novel therapeutic strategies by identifying critical regions that influence proinsulin stability.” could be added. Direction for future research is missing. “These findings need to be experimentally validated and tested in an in vivo context.”
This article makes important contributions to our understanding of the molecular effects of diabetes-associated mutations by examining in detail the role of Arg(B22) in proinsulin stability. Powerful computational methods such as AlphaFold and metadynamics were used, but it should be emphasized that the results should be supported by experimental validation. In terms of grammar, there are some minor grammatical errors and phrases that affect fluency, and the justification of the parameters used in the methodology section could be more detailed. Overall, although the study is built on a solid scientific foundation, a better discussion of the clinical implications of the findings and their place in the literature would be useful.
Author Response
The authors investigated the role of Arg(B22) in proinsulin stability by AlphaFold predictions and metadynamic simulations. They showed that Arg(B22) maintains structure by interacting with Asn86, but replacement with Gln or Glu disrupts this interaction, increasing conformational flexibility. The R(B22)Q variant favors a more unfolded state, while other mutations destabilize proinsulin by weakening hydrogen bonds. This study explains at the molecular level how R(B22) mutations disrupt proinsulin structure and its association with diabetes. However, authors should make some corrections throughout the text.
- An appropriate form of citation style should be applied based on the biomolecules. Please give a reference for lines 36 to 38. “including Gly, Ala, Lys, Tyr and Phe”
Response: We have followed to Biomolecules free formatting, and accordingly our citation style includes authors, article title, and journal information including issue and page numbers.
Based on the reviewer’s suggestion, we now consistently stick to a 1-letter amino acid code.
- In this sentence, in line 16, the comma is in the wrong position. It should be before “and”.
Response: We have incorporated this change.
- The sentence “The dynamics of the protein therefore become crucial to its function.” is on line 99 and 'therefore' should be separated by two commas; 'becomes' should be 'become' instead of 'becomes'.
Response: We have now reworded the sentence accordingly.
- The sentence between lines 46 and 48 should be made more fluent.
Response: We have now reworded the sentence accordingly for better readability
- The sentence between lines 42 and 43 should be improved by adding “which was”.
Response: We have now reworded the sentence accordingly.
- In line 79, Latin phrases such as "in silico" should be italicized.
Response: All instances of in silico, in vitro, etc. have been italicized.
- Some sentences contain extra spaces. Please revise them (e.g., line 81).
Response: We have now corrected this.
- In lines 95-97, writing the number as “eight” instead of “8” is more in line with academic writing rules, and it is more correct to use “:” instead of “ - ”.
Response: We have now reworded the sentence accordingly.
- Lines 100-102 should be “AlphaFold-predicted” and “minimum-energy” should be combined.
Response: We have now reworded the sentence accordingly.
- In lines 136-138, “observe” instead of “see” is more appropriate in academic writing.
Response: We have now reworded the sentence accordingly.
- In lines 338-339, “a certain” instead of “some” sounds more academic.
Response: We have now reworded the sentence accordingly.
- In line 344, the phrase "(CITE Some Experiment if available)" appears. Please remove it.
Response: Thanks. We have removed it.
- In lines 356-358, “results in” is more precise than “leading to” and “flexible” is more common than “malleable”.
Response: We now use flexible instead of malleable in the revised text.
- For the abstract section, it would be helpful to make the methods more explicit. For example, why AlphaFold and metadynamics are preferred. Also, to emphasize the association with diabetes more strongly: “These findings provide important contributions to understanding the molecular mechanisms of diabetes-associated proinsulin mutations.”
Response: Thank you. We now mention this in the revised abstract.
- The introduction could better contextualize why R(B22) mutations are specifically important and previous studies on this topic.
Response: Thank you. We have now included evidence from five studies in the introduction emphasizing the importance of R(B22) in proinsulin.
- A comparison with other insulin mutations associated with diabetes could be included. More explanation on the pathophysiology of the mutations should be added.
Response: Thank you for the feedback. We have incorporated the comparisons to other insulin mutations and expanded on the pathophysiology in the Discussion section.
- For the methodology part, it should be more clearly explained why the simulation parameters were chosen based on the literature.
Response: We now include a more detailed description of the simulation methods in the revised manuscript.
- For the conclusion, more details on the statistical significance of the results should be provided. The question “How do the results differ from other previously identified insulin mutations?” should be answered.
Response: We acknowledge the referee’s concerns. The structures were computed based on an extensive sampling protocol using the metadynamics algorithm. Unlike conventional MD simulations, we ensure that the system samples all the states during the trajectory while computing the free energy profiles in the paper. The delta_G for the proteins in their open and closed configurations are of the order of > 10 kcal/mol for proteins that favor the closed state. On the other hand, for destabilized mutants like Arg22Gln, the delta_G between the open and closed state minima is < 5 kcal/mol suggesting a significant difference. As per our knowledge, for R(B22) mutants, these are the first reported MD simulation results making a comparison with previous literature difficult.
- Figures (e.g., Fig. 1 and 2) need more explanatory discussion. The captions of the figures should be made more descriptive.
Response: We now include a more detailed caption.
- For the discussion part, the implications of the findings for clinical applications should be discussed further. A topic such as “Do these mutations have the potential to be considered as therapeutic targets in the future?” could be addressed. Limitations of the study (e.g., lack of in vitro/in vivo validation) should be clearly stated.
Response: Thank you for your suggestions. We've addressed the study's limitations in a newly added final paragraph in the Discussion section. Also, future directions, including possible therapeutic avenues are now included in the Conclusion and Future Directions section.
- In the overall conclusion, the original contribution of the study should be clarified. A statement such as “This study sheds light on novel therapeutic strategies by identifying critical regions that influence proinsulin stability.” could be added. Direction for future research is missing. “These findings need to be experimentally validated and tested in an in vivo context.”
Response: We now include this in the “Conclusions and Future Directions” section of the revised paper.

Reviewer 2 Report
Comments and Suggestions for Authors
In the manuscript “ The B22 dilemma: structural basis for conformational differences in proinsulin B-chain Arg22 mutants” the authors explored the role of R(B22) in proinsulin stability, using AlphaFold-predicted structures and metadynamics simulations. The manuscript is well written however there are several flaws in the manuscript which need to be addressed before considering it for publication.
- Throughout the main text, especially in the abstract, both three letter and one letter codes are used to represent various amino acids. It is confusing for the readers and I haven’t seen this kind of mixed usage. For example lines 10-14. It should be corrected throughout the text.
- Authors used AlphaFold 3 for the structure prediction. How do the structures differ with respect to AlphaFold 2? Are there any advantages for AlphaFold 3?
- It is not clear for the readers whether stimulation of AlphaFold prediction provided structures shown in Figure 2. Provide a detailed figure legend and adjust the main text accordingly.
- Although ensemble averaged structure is shown, I would like to see the full ensemble and how unique the conformational sampling is.
- If AlphaFold 3 is used for structure prediction, provide the other four structures as supplementary. The manuscript should be detailed enough with all necessary data collected.
The above-mentioned comments are representative review comments/questions. Thus there is room for improvement and I would like to hear from the authors before considering the manuscript for publication.
Author Response
In the manuscript “ The B22 dilemma: structural basis for conformational differences in proinsulin B-chain Arg22 mutants” the authors explored the role of R(B22) in proinsulin stability, using AlphaFold-predicted structures and metadynamics simulations. The manuscript is well written however there are several flaws in the manuscript which need to be addressed before considering it for publication.
- Throughout the main text, especially in the abstract, both three letter and one letter codes are used to represent various amino acids. It is confusing for the readers and I haven’t seen this kind of mixed usage. For example lines 10-14. It should be corrected throughout the text.
Response: We now consistently stick to a 1-letter amino acid code.
- Authors used AlphaFold 3 for the structure prediction. How do the structures differ with respect to AlphaFold 2? Are there any advantages for AlphaFold 3?
Response: The major difference between AlphaFold2 and AlphaFold3 is in the area of multimeric models such as protein-protein interaction and protein-ligand interaction predictions. Since we employed these models as a starting point for single molecule simulations, the choice of AlphaFold2 vs AlphaFold3 is not significant. Furthermore, we employ an enhanced sampling protocol to ensure that we extensively study the free energy landscape of the protein.
- It is not clear for the readers whether stimulation of AlphaFold prediction provided structures shown in Figure 2. Provide a detailed figure legend and adjust the main text accordingly.
Response: Fig 2 now has simulation ensemble-averaged structures from the simulation. We mention this in the revised figure legend.
- Although ensemble averaged structure is shown, I would like to see the full ensemble and how unique the conformational sampling is.
Response: We now introduce a new Figure 3 wherein the WT minimum energy configuration is compared with the corresponding ensemble-averaged structures from R(B22)E and R(B22)Q. We also introduce a new Figure 7 wherein we show the ensemble-averaged configurations that correspond to the open and closed states, for every variant under study in this paper.
- If AlphaFold 3 is used for structure prediction, provide the other four structures as supplementary. The manuscript should be detailed enough with all necessary data collected.
Response: AlphaFold3 structures for all models are provided as a GitHub repository link in Materials and Methods, which may also be accessed here: https://github.com/sranga88/Proinsulin-R-B22--AF3-Models.git.
The above-mentioned comments are representative review comments/questions. Thus there is room for improvement and I would like to hear from the authors before considering the manuscript for publication.
Reviewer 3 Report
Comments and Suggestions for Authors
See attached file.

Author Response
The B22 dilemma: structural basis for conformational differences in proinsulin B-chain Arg22 mutants, by Ranganathan and Arunagiri
This paper describes the investigation of the predicted structural differences among proinsulins with various mutations at the B22 position, with particular emphasis on the differences between R(B22)E and R(B22)Q. AlphaFold was used to provide starting structures for each of the mutants, and metadynamics was used to examine the free energy landscape along a reaction coordinate represented by Rg of the molecule. The method appears to be sound and the results reasonable and informative. Some things in the presentation should be improved for clarity and consistency:
- For those not so familiar with proinsulin, it would be useful to explain how the 86-residue sequence maps to the A, B, and C sequences. Perhaps a small sketch could be added to Figure 4.
Response: Thanks for the suggestion. We have now included the said figure in Fig 1 of the revised manuscript. Also, in a small introductory paragraph at the start of the Results section, we clarify the nomenclature and the various ways of referring to proinsulin amino acid residue locations and substitutions.
- The designation of residues is inconsistent and confusing; sometimes the position in proinsulin is specified and sometimes the position within an individual chain. For example, an important interaction is referred to as "R(B22)-Asn86". Why not "Arg22-Asn86", or "R(B22)-N(A21)"? On line 36, the mutated residue is described as "position 46 in proinsulin", but this presumably should be "position 46 in preproinsulin".
Response: Thank you for pointing this out. We have now fixed this for better readiblility and also introduced a paragraph at the start of the Results section we clarify the nomenclature and the various ways of referring to proinsulin amino acid residue locations and substitutions.
- The listing of mutants investigated is inconsistent: on lines 97, 346, and 348 the last one is given as Pro, but on line 226 and in figures 5-7 it is Asp. The abstract doesn't mention either Pro or Asp. Line 123 says there are 9 mutants, rather than the 8 mentioned everywhere else.
Response: We now correct these errors in the revised text. Indeed, 8 mutants were studied instead of 9 as erroneously reported in line 123 previously. Also, Asp is now mentioned in the revised abstract, and the incorrect reference to Pro is now removed. We thank the referee for this observation.
- Figure 2, although attractive, could be improved. The objects made up of green balls represent, I think, sidechains involved in hydrogen bonding, but a stick representation would be clearer, or perhaps just circles for the H-bonds themselves, rather than showing sidechains. There is no explanation of what the object composed of gray balls is. Judging from the other figures, there should still be some Hbonds with "highest involvement", whatever that means, in panel C, but none are shown.
Response: We have now modified this figure (now Fig 1 B, C, D) to show the H-bonds in green-colored CPK ball and stick representation for the residues that show the highest involvement in inter-domain interactions. As rightly pointed out, the figure corresponding to R(B22)Q is now corrected to show the residues involved in frequent interactions.
- Figures 4 and 7 are useful for pointing out the difference in number of A-B, A-C, and B-C H-bonds among the different mutants, but they could also be improved. The lettering is tiny, and in Fig. 4 the labels on the x-axis and the color scale are cut off. The color scale is labeled "Hydrogen bond occupancy" - what does this mean? It's probably not the "occupancy" reported in experimentally determined crystal structures. It might be mentioned that the red, blue, and green boxes only surround half of the potential interactions between each pair of chains; for example, the red box shows interactions where residues in the A chain are donors and those in the B chain are acceptors, but there is also the reverse case. The caption could also mention that the significance of an interaction is shown by both the color and the size of the circle.
Response: Yes, as rightly pointed out by the reviewer, occupancy here is different from that reported in crystal structures. By occupancy, a quantity shown in %, we mean the frequency with which an amino-acid residue pair is involved in an interaction in the simulation. For instance, an occupancy of 30% would mean that the two residues are within an interaction cutoff threshold for 30% of the frames corresponding to the free energy minimum population. The red, blue and green boxes are only shown as a visual aid to highlight the part of the map that shows the greatest difference in interactions, hence only shown along one axis even though the referee correctly points out the asymmetry of the map. We now introduce this explanation in the figure legend of the revised text.
We understand the concern regarding the font size in Fig 4 and 7. However, due to the dense nature of the map with several interactions nearby, a larger font size results in the lettering being obscured by nearby interaction labels. Due to the constraints of the journal’s word template, the figures are resized to fit in. However, we do have higher-resolution images with increased readability which will be made available during the publication stage.
- On lines 211 and 344, please supply actual references.
Response: We have provided the references (now lines 251, 468 and 469 in the revised manuscript)
- There seems to be something missing on line 203 between "across species13,14." and "is vital part".
Response: We have now corrected the sentence.
- On line 351, B24 is mentioned for the first time; I suggest adding a reference here, modifying the text to "is highly susceptible to substitutions in B22, as it is to changes in B2416, and".
Response: We now reword the statement as suggested, and also cite a reference for B24.
- On line 370, "the edge of their foldability" is mentioned. It's not immediately obvious to me what this means; suggest adding a reference to explain the phrase (ref. 16).
Response: We now add a line explaining what “edge of foldability” means with respect to the folding landscape of the protein.
- In the Author Contributions, omit instructions "For research articles...should be used".
Response: We have removed the instructions
- 15 is missing the journal name.
Response: The said reference (Landreh, et al.) is now corrected (see Ref 23 in the revised manuscript)
- The English is very good, but there are a few typos:
Line 222: "Fig 5 and F" should be "Figs 5B and 7B".
Line 235: "peptid" should be "peptide".
Line 256: "minia" should be "minimum".
Line 278: "B-Chan" should be "B-chain".
Response: We have now corrected the above in the revised manuscript.
Reviewer 4 Report
Comments and Suggestions for Authors
The authors of this manuscript analyze the effect of so-called B22 mutation of insulin B-chain using AlphaFold for 3D structure predictions of proinsulin mutants followed by metadynamics simulations. The authors conclude that the mutants are destabilized compared with wild type proinsulin, showing larger Rg and loss of hydrogen bonds. B22 mutation has been confirmed as a crucial point for interaction of insulin with the receptor. Thus, the topic of the present manuscript is interesting and significant and this manuscript is worth publishing but the following points should be reconsidered before publication.
- The authors show the decreases of hydrogen bonds in mutants presenting contact maps of contact maps. But, some of readers would like to see the details of molecular basis of the changes of hydrogen bonds, that is, it is necessary to show the 3D structures of figures presenting changes of main chain and side chains between wild type and mutant insulins. There are sentences in page 4, lines 152-178, “To further understand the molecular basis for the relatively lower stability of the native state in the variants compared to the WT, we first computed the number of stable inter-chain hydrogen bonds in the WT with that of R(B22)E and R(B22)Q….. Similarly, C-peptide and A-chain interact via Tyr84-Arg65 and Ser77-Leu58 as the primary contacts along with a set of dynamic interactions that could form and break constantly (smaller, darker circles in Fig 4A, red box). Overall, the prevalence of several inter-chain interactions between A-chain-B-chain and A-chain-C177 peptide helps stabilize the WT proinsulin in the more compact configurations (Fig 1A and 2A).”. This part should be reorganized using the figures of protein 3D structures.
- The protocol of metadynamics should be described in detail including free energy calculation.
- Relating to the comment 1, in page 5, lines 217-229, the authors mention that several mutants show the open configurations except R(B22)D, more wild-type like configuration. What is the reason of this results? The authors should explain this result on molecular basis, that is, change of 3D structures of main chain and side chains.
Minor point;
In page 3, line 112, what is “CV”?
Author Response
The authors of this manuscript analyze the effect of so-called B22 mutation of insulin B-chain using AlphaFold for 3D structure predictions of proinsulin mutants followed by metadynamics simulations. The authors conclude that the mutants are destabilized compared with wild type proinsulin, showing larger Rg and loss of hydrogen bonds. B22 mutation has been confirmed as a crucial point for interaction of insulin with the receptor. Thus, the topic of the present manuscript is interesting and significant and this manuscript is worth publishing but the following points should be reconsidered before publication.
- The authors show the decreases of hydrogen bonds in mutants presenting contact maps of contact maps. But, some of readers would like to see the details of molecular basis of the changes of hydrogen bonds, that is, it is necessary to show the 3D structures of figures presenting changes of main chain and side chains between wild type and mutant insulins. There are sentences in page 4, lines 152-178, “To further understand the molecular basis for the relatively lower stability of the native state in the variants compared to the WT, we first computed the number of stable inter-chain hydrogen bonds in the WT with that of R(B22)E and R(B22)Q….. Similarly, C-peptide and A-chain interact via Tyr84-Arg65 and Ser77-Leu58 as the primary contacts along with a set of dynamic interactions that could form and break constantly (smaller, darker circles in Fig 4A, red box). Overall, the prevalence of several inter-chain interactions between A-chain-B-chain and A-chain-C177 peptide helps stabilize the WT proinsulin in the more compact configurations (Fig 1A and 2A).”. This part should be reorganized using the figures of protein 3D structures.
Response: We thank the referee for the suggestion. In the revised text, we now add a new figure where we show superimposed structures of R(B22)E and R(B22)Q aligned onto the WT proinsulin 3D structure. These structures are ensemble averaged structures from the simulation corresponding to the free energy minima. We hope that the new figure (Fig 3 in revised text) can act as a visual aid for the readers to understand the conformational divergence of these structures upon mutating the Arg22 residue. Interestingly, the C-peptide in both variants shows significant structure development, a feature that is absent in the WT proinsulin.
- The protocol of metadynamics should be described in detail including free energy calculation.
Response: We mention this in the Methods section of the revised manuscript.
- Relating to the comment 1, in page 5, lines 217-229, the authors mention that several mutants show the open configurations except R(B22)D, more wild-type like configuration. What is the reason of this results? The authors should explain this result on molecular basis, that is, change of 3D structures of main chain and side chains.
Response: We thank the referee for the comment. While all R(B22) variants show a drop in A-chain-B-chain interactions, the variants gain in A-chain-C-peptide and B-chain-C-peptide interactions. The two variants across our study which show the highest degree of destabilization, R(B22)Q and R(B22)Y, show the lowest B-C interactions while also losing out on A-B interactions which stabilize the WT. All other variants lie somewhere in between the WT and these extremely destabilized variants – R(B22)Y and R(B22)Q. Interestingly, the R(B22)D does show a free energy profile closer to the WT. An R->D mutation is one of the known substitutions evolutionarily in proinsulin at this location (Figure 1 from Landreh, et al. BioMol Concepts 2014; 5(2): 109–118). A potential cause of the relative stabilization of this variant could be attributed to a cluster of interactions between the A-chain and C-peptide, for this variant. We now discuss this in the revised text.
Sequence alignment of proinsulin sequences from several species. https://doi.org/10.1515/bmc-2014-0005
Minor point: In page 3, line 112, what is “CV”?
CV = Collective Variable

Round 2
Reviewer 1 Report
Comments and Suggestions for Authors
The authors have successfully completed the revision process. In this form, the article is suitable for publication.
Author Response
We are glad that the reviewer is satisfied with our revision. The authors would like to thank the reviewer for their valuable feedback and suggestions.
Reviewer 2 Report
Comments and Suggestions for Authors
By addressing all the comments, the manuscript is significantly improved. I support publishing it in Biomolecules, after proofreading.
Author Response
The authors have addressed most of the reviewers' concerns, and have improved their manuscript. However, numerous minor issues remain:
1) The Introduction should be expanded to provide an overview of the main results and of their potential impact.
Response: Thank you. We have now added a paragraph briefly describing the main results at the end of the introduction.
2) The authors should add more details in terms of the input provided to AlphaFold for the different model predictions. At this stage it could be useful to refer to Figure 1A.
Response: In the Materials and Methods section, we have included all the primary sequences we used as the input for AlphaFold structure prediction. Figure 1A has been referred to here as suggested by the reviewer.
3) The authors should provide plots showing convergence of the metadynamics simulations as supplementary materials.
Response: Plots showing metadynamics convergence has been included in the supplemental file (Figure S1).
4) In Figure 1, it is unfortunate that the blue-red color code for both chains is not the same in subplot A as in subplots B,C,D. This should be fixed. The authors should also indicate the location of Asn86 in subplot A, which would help understand the important interaction with R(B22) they mention in the text (this would also help with the different naming conventions).
Response: We thank the reviewer for pointing out this. The colors have been changed to be consistent across all panels.
5) In its current form, Figure 5 is unreadable. The subplots should be displayed in a vertical layout, instead of a horizontal layout, or alternatively, over two rows.
Response: As suggested, the figure has now been enlarged.
6) In its current form, Figure 9 is also unreadable. The subplots should be displayed in a 3x2 layout, instead of a 2x3 layout.
Response: As suggested, we have now replaced the 2x3 layout with a 3x2 layout.
7) The authors were asked to improve the consistency of amino acid notations (i.e., one-letter vs three-letter codes). However, some inconsistencies remain, as both codes are mixed up in several places:
- in the abstract: Arg(B22)Q -> R(B22)Q
- in the abstract: Glu or Q -> E or Q OR Glu or Gln
- in the introduction: B22(Arg) to Glutamine or Glutamic Acid -> B22(Arg) to Gln or Glu
- in the caption of Figure 2: R22E -> R(B22)E
- in the titles of sections 3.3 and 3.4: Arg(B22)-variants -> R(B22)-variants
- both Asn86 and N86 are used throughout, when a single notation should be used instead
- in figure 7: Arg22Q -> R(B22)Q
- in figure 8: R(B22)Tyr -> R(B22)Y
- T30-Asn83
- Asp-substitution -> D-substitution etc.
Response: We have fixed the inconsistencies now.
8) Typos and errors remain, such as:
- in the caption of Figure 2: WT-like native minia -> WT-like native minimum
- in the caption of Figure 2: A defined minima -> A defined minimum
- in the caption of Figure 2: no stable minima -> no stable minimum
- in section 3.2: contacts within the blue box in Fig.4A -> contacts within the blue box in Fig 5A
- in section 3.2: R(B22)->Q22 -> R(B22)Q
- in section 3.2: several different proinsulin structures corresponding to different species is vital -> several proinsulin structures corresponding to different species, and is a vital
- in the caption of figure 5: B-Chan -> B-Chain
- in the caption of figure 5: spheres -> circles
- in sections 3.3 and 3.4: the various instances where "configuration" is used should be replaced by "conformation"
- in section 3.4: B-chain-C-peptid -> B-chain-C-peptide
- in the caption of Figure 9: Fig 4A -> Fig 5A
- in the caption of figure 9: spheres -> circles
- in the Discussion, at the top of page 13: malleable -> flexible
Response: We thank the reviewer for pointing out our mistakes. These have been corrected in the revised draft.
